# Sustainability Orientation and Focus in Logistics and Supply Chains

**Sebastjan Lazar** [1], **Dorota Klimecka-Tatar** [2] **and Matevz Obrecht** [1,*]

1    Faculty of Logistics, University of Maribor, Mariborska Cesta 7, 3000 Celje, Slovenia; sebo.pro@gmail.com
2    Faculty of Management, Czestochowa University of Technology, al. Armii Krajowej 19b,
      42-200 Czestochowa, Poland; d.klimecka-tatar@pcz.pl
*    Correspondence: matevz.obrecht@um.si; Tel.: +386-03-428-53-67

**Abstract:** Sustainable development, logistics, and supply chain are being combined into three increasingly connected and topical global research areas. Therefore, this paper's novelty identifies and defines the priorities of the UN Sustainable Development Goals and sustainable development dimensions in supply-chain- and logistics-management-related studies in the last decade. Knowing logistics and supply chain sustainability focus and orientation is valuable information for researchers and managers to adapt and mitigate their business logistics according to the forecasted trends. The paper provides a systematic and comprehensive review of the literature and is based on 116 scientific papers published between 2010 and 2020 in 73 international journals in the Scopus and Web of Science databases. The study defines focus with integrating environmental, social, and economic sustainability for logistics- and supply-chain-related studies. It emphasizes primary and secondary links of investigated studies with 17 United Nations sustainable development goals. The bibliometric analysis also examined keyword relations. One of the main contributions is that economic sustainability was identified as the most represented one-dimensional sustainability focus. It was revealed that supply chain studies integrated all three sustainability dimensions more frequently (50.60%) than logistics studies, which were equally related to studying two- or three-dimensions of sustainability (39.39%). The most significant findings are also that studies were identified to be oriented primarily towards "responsible consumption and production", "industry, innovation, and infrastructure" and "affordable and clean energy" and secondary especially on "sustainable cities and communities".

**Keywords:** sustainable development (SD); sustainable development dimensions; sustainable development goals (SDG); logistics; supply chain management

## 1. Introduction

### 1.1. Background

The environment around the world needs to be protected to ensure economic and social development for the benefit of different generations [1]. This is only one of the reasons why sustainable development has gained worldwide attention in recent years [2] and its role is still growing [3], enabling us to solve current and future fundamental challenges of humanity [4] not only for the sustainability of natural resources and the environment but also the sustainable well-being of people and the happiness of the population [5]. Sustainable development has become a very popular slogan of modern development discourse [6] and since the emergence of the concept in the 1980s [7] when it was defined as a way of meeting the needs of the current population without jeopardizing the ability of successors to meet their needs. It now appears in almost every sector of our life and economy [8]. Shortly after the mentioned definition, it was found [9] that many non-governmental and also government organizations have accepted sustainable development as a kind of new development paradigm.

### 1.2. Literature Review

Many scientists [10–13] warning that current climate changes, which are one of the main threats of the modern age, additionally empowers its implementation. One of the primary reasons for this is the high global dependence on fossil fuels for production, transport, etc. [14] with logistics and supply chains as an integral part. Some scientists [15] highlight that cleaner modes of transportation, sustainable production, logistics, and supply need to be developed to address the challenge of increasing population and limited natural resources. The authors of [16] primarily emphasize the confrontation of supply chains with environmental sustainability. The authors of [17] exposed that governments worldwide have introduced regulations for industries to produce sustainable items, consequently increasing the demand for environmentally friendly materials in various industries [18]. It was also mentioned [19] that companies are increasingly adopting a strategy that includes a positive sustainability aspect, for example, with various inputs in the production process and logistics and supply chain operations [20]. The importance of sustainable development in companies is also mentioned [21] as it is expected to become one of the critical conditions for international competitiveness even in the strongly related field of supply chains and logistics. There are many different definitions for both areas. One of many supply chain definitions [22] says that a supply chain is a system of activities (cooperating organizations) dealing with design, coordination, and controlling of materials, components, and finished products from the supplier to end customer and back in case of wastes. On the other side, it was, for example, mentioned [23] that in 1991, the Logistics Management Council defined logistics as the process of planning, implementing, and controlling the efficiency of the flow and storage of goods, services, and related information from the company to the consumption of them and is, therefore, more oriented on one company. Some of the scientists [24] state that both logistics and supply chain, through previous and current development, have moved to the level of an important business function in companies, which further confirms that research about sustainable development in this area increases its importance. There are three dimensions to sustainable development: social, economic, and environmental [25]. There are also 17 sustainable development goals (SDG), defined by the United Nations 2030 Agenda for Sustainable Development [26] with the aim and goal of achieving a better and more sustainable future. Sustainability dimensions and SDG are becoming more and more relevant to be incorporated in business logistics and supply chain management. Due to limited resources in organizations, such transition priorities must be studied so that they enable appropriate business decisions on different levels of national and international organizations.

### 1.3. Research Gap, Question, and Goals

It was found out that current studies concerning the three research areas are dealing with separate partial issues only, for example: how far customers of logistics services are willing to consider aspects of sustainable development [27], the logistics service provider's approach to supporting SDGs [28], sustainable development through waste recycling in supply chains [29], and sustainable development in global supply chains [30]. Studies on a systematic review of the literature in the indicated fields were detected only in, for example, the following areas: performance measurements and metrics in logistics and supply chain management in the specified period [31], understanding the role of logistics capacity in achieving supply chain flexibility [32], green supply chain management [33], theories in the field of sustainable supply chain management [34], and research of supply chain with petroleum and biofuels [35]. With respect to reviewed literature, lack of studies in a systematic review of research trends in sustainable development in general but especially sustainable development perspectives and goals in logistics and supply chains was detected. The research gap was also identified in examining research studies regarding their focus on specific sustainable development dimensions—economic, social, and environmental as well as their combinations to see whether, for example, environmental sustainability is ranked higher than, for example, social sustainability. Therefore, this paper aims to review

and analyze sustainability orientation according to SDG and define sustainability focus for studies published from 2010 to 2020 (until the beginning of October).

Based on the gap, the following research questions were investigated in this study: (1) Which of the three SD dimensions are integrated into logistics- and supply-chain-related studies and to what extent? (2) Which SDG priorities can be related to logistics- and supply-chain-related studies and which are of secondary importance? (3) What are the differences between logistics and supply chain-related studies in sustainability orientation and focus?

The goal of this research is to reveal research relations of sustainability in logistics- and supply-chain-related studies separately since supply chain is more related to cooperation among numerous parties in the supply chain and logistics is more pertaining to activities within a single organization. This research focuses on analyzing primary and secondary research orientation in relation to 17 SDGs regarding international recognition and devotion of science to SDGs. The contribution of the research is to reduce the gap and provide new insights into sustainability orientation and focus in logistics and supply chain research, which also reflects the future of business logistics.

The study is structured as follows: (a) materials and methods; (b) results section, divided into sustainability perspective focus, sustainability orientation related to SDG and constructing, visualizing, and analyzing bibliometric networks of studied papers with VOS viewer; (c) discussion and interpretation; and concluding remarks in (d) conclusion.

## 2. Materials and Methods

In preparing the research, the methodology of conducting a systematic literature review was constructive [36]. This paper provides a systematic and comprehensive review of available literature on logistics and supply chains with a focus on their relations with sustainable development. This research included published papers in Web of Science and Scopus databases from 2010 to October 2020. Mentioned bases were used because they are most often used (in comparison to other bases) and at the same time differ in coverage [37]. Consequently, the use of both databases expands the quality of research and findings. After reviewing different combinations of terms and keywords, it can be concluded that the focus of research is most covered if the search is by title. The paper selection process and research process are presented in Figure 1.

A database of studied material and analysis represented by 116 references [16,20,27–30,38–147] are included in the Supplementary Materials which is submitted as a separate file due to too large a volume of data. Papers from the Supplementary Materials were analyzed according to:

- The geographical area of research, to investigate national vs. global studies;
- Three dimensions of sustainable development: social, economic, and environmental [25] to investigate which dimension is most covered and how well sustainable development in logistics and supply chain is investigated from all three perspectives;
- The 17 SGDs defined by the United Nations (for achieving a better and more sustainable future): (1) No poverty, (2) Zero hunger, (3) Good health and well-being for people, (4) Quality education, (5) Gender equality, (6) Clean water and sanitation, (7) Affordable and clean energy, (8) Decent work and economic growth, (9) Industry, Innovation, and Infrastructure, (10) Reducing inequalities, (11) Sustainable cities and communities, (12) Responsible consumption and production, (13) Climate actions, (14) Life below water, (15) Life on land, (16) Peace, justice and strong institutions, (17) Partnerships for the goals [26] with the aim and goal to define priorities in research studies orientation. According to the mentioned goals of sustainable development, the research was divided into primary and secondary perspectives. If the research was strongly related and focused on a specific goal, it was considered as the primary perspective. Otherwise (if for example, only partly related to one SDG) then as a secondary perspective.
- Main findings to identify the research emphasis application.

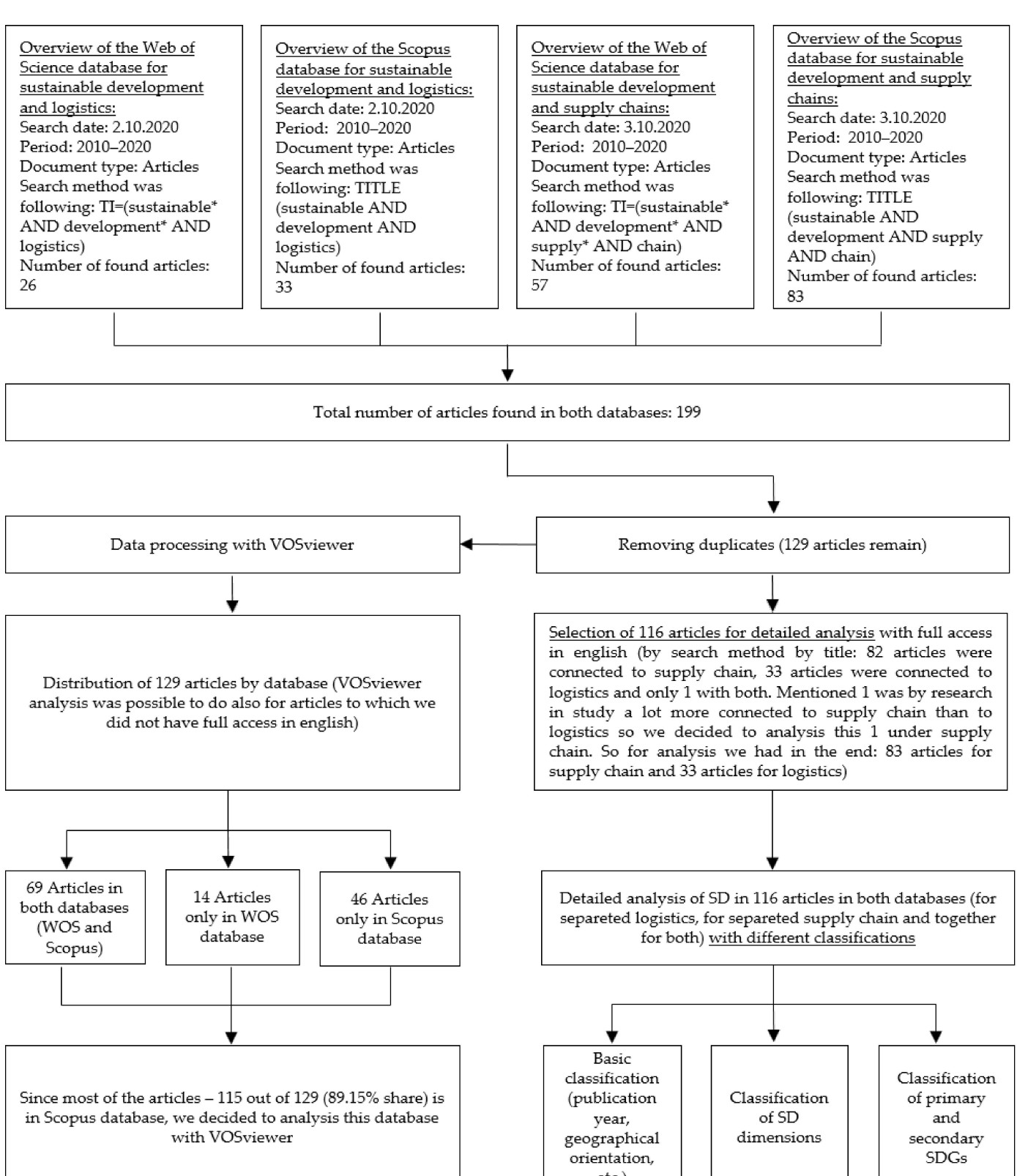

**Figure 1.** Paper identification and selection process within the research process.

Further, the research focuses on constructing and visualizing bibliometric networks with VOSviewer, enabling a wide variety of bibliometric analyses [148–150]. In this case, results of network connectivity (sustainable development, logistics, and supply chain) were investigated as well as citations to assess the most influential research studies.

After detailed data processing, the results were analyzed and interpreted in the following structure:

- Relations to journals of publication.
- Focus on separate sustainable development dimensions.
- Priority orientation of investigated studies in relation to SDGs.
- Secondary orientation of investigated studies related to SDGs.
- Bibliometric analysis and network connectivity based on VOSviewer.

## 3. Results

### 3.1. Relations to Journals of Publication

In this article, 116 studies published in 73 international journals in the period between 2010 and 2020 were taken into consideration. In the year 2020, only studies published from January to the beginning of October were taken into consideration. According to the geographical orientation, there were 29 international studies, 63 national, and the rest (24) of the studies were not specified. The fact which is interesting is that in the years 2019 and 2020, there is a much bigger focus on international studies than in the previous period range (48.28% of all international studies were published between these two years). Most studies related to sustainability issues related to logistics and supply chain were published in 2019, namely 29, representing 25% of all studies. A total of 74 out of 116 studies (63.79%) were published in 2018, 2019, and 2020 which shows that this research field is gaining scientific attention and is really topical. Published studies by year and scope are shown in Figure 2.

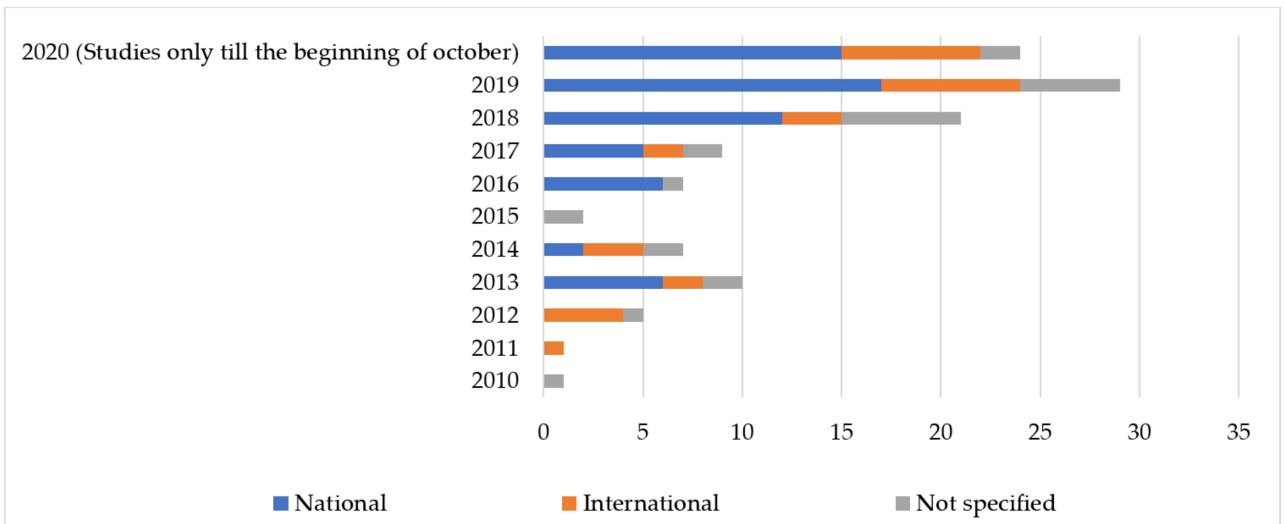

**Figure 2.** Number of published studies by year and scope.

Analyzed studies were most present in four international journals: International Journal of Supply Chain Management, Sustainability, Journal of Cleaner Production and Business Strategy and the Environment (30.17% of studies). Distribution among journals can be seen in Table 1.

Out of 116 articles, 11 were included in Web of Science, 37 in Scopus, and 68 were included both in the Web of Science and Scopus databases.

**Table 1.** Number of published studies by journals.

| No. | Studies Published in a Journal | Number of Studies |
|---|---|---|
| 1 | International Journal of Supply Chain Management | 12 |
| 2 | Sustainability | 9 |
| 3 | Journal of Cleaner Production | 8 |
| 4 | Business Strategy and the Environment | 6 |
| 5 | Corporate Social Responsibility and Environmental Management | 3 |
| 6 | Environmental Science and Pollution Research | 3 |
| 7 | International Journal of Productivity and Performance Management | 3 |
| 8 | Sustainable Development | 3 |
| 9 | Ecological Economics | 2 |
| 10 | Flexible Services and Manufacturing Journal | 2 |
| 11 | Greening of Industry Networks Studies | 2 |
| 12 | Polish Journal of Management Studies | 2 |
| 13 | Other journals in which only one study was published | 61 |
| | Total | 116 |

### 3.2. Focus on Sustainable Development Dimensions

To define the focus of sustainability perspectives, different partial sustainability dimensions were analyzed. At least one dimension of sustainability (separately, in a combination of two or all three dimensions together) was identified 266 times meaning that most of the papers included multiple sustainability dimensions. In total, 55 studies (47.41% of all) were related to all three dimensions; 34.48% were a combination of two dimensions—40 studies and 21 studies (18.10%) were related to one sustainability dimension only. Most represented was the economic dimension, defined in 97 cases (83.62% of all studies), followed by the environmental dimension in 90 cases (77.59% of all studies) and social dimension in 79 cases (68.10% of all studies).

Figure 3 clearly shows that supply-chain-related research is more focused on integrating all three dimensions of sustainability than those related to logistics. Within "logistics" related studies, 13 were identified to be related with all three dimensions, 13 with two dimensions (39.39% of all), and 7 studies with one dimension only (21.21% of all). In supply-chain-related studies, 42 studies (50.60% of all) were identified to be related to all three dimensions. 27 studies (32.53% of all) were related to a combination of two dimensions, and 14 studies (16.87% of all) with one separate dimension only.

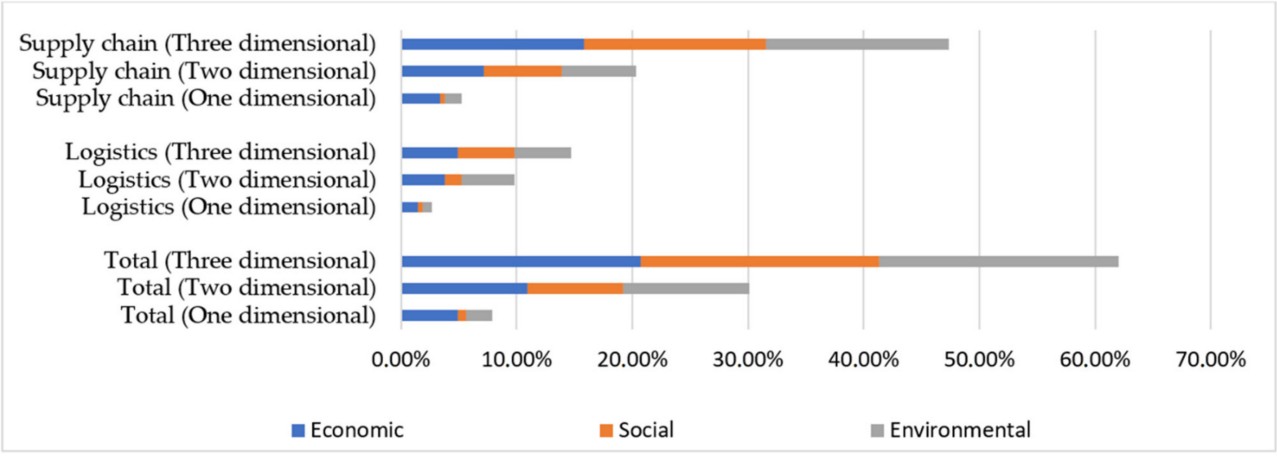

**Figure 3.** Frequency of studies with identified one-, two- or three-dimensional integration of Sustainable Development dimensions.

### 3.3. Priority Orientation of Investigated Studies towards Specific SDGs

Among the 17 SDGs, primary orientation was defined 193 times for all 116 studies since some studies' primary orientation can be related to multiple SDGs. A separate analysis of results was made for logistics (33 studies—41 SGD relations) and supply chain (83 studies—152 SDG relations). In Figure 4, the primary orientation towards specific SDGs is presented. Presented data were analyzed (1) as a relation to one SDG only (light blue—logistics and light green—supply chain) or (2) as a relation to multiple SDGs (dark blue—logistics and dark green—supply chain). In total, six goals were defined as being the top priority, out of which three: "Responsible consumption and production" (especially for supply chain perspective), "Industry, Innovation, and Infrastructure", and "Affordable and clean energy" stand out. These three appeared 27 times. The other three: "Reducing inequalities" (again especially important for supply chain perspective), "Decent work and economic growth" (entirely related with supply chain perspective), and "Partnerships for the goals" appeared in 22 cases. Among logistics studies, primary orientation towards "Industry, Innovation, and Infrastructure" (appeared 10 times) and "Affordable and clean energy" (appeared 9 times) stands out. In the primary orientation of supply chain studies, two SDGs stand out but not the same SDGs as in the case of logistics studies. Here, "Responsible consumption and production" and "Decent work and economic growth" were identified 22 times. In total, 14 studies (12.07% of all studies) are primarily related to "Industry, Innovation, and Infrastructure," which is the most represented SDG among studies with an orientation towards a single SDG. Among studies with single SDG orientation, these were followed by: "Partnerships for the goals" (in a total of 12 out of 116—10.34% of all studies) and "Responsible consumption and production" (in a total of 11 out of 116—9.48% of all studies). In logistics-related studies, it was found out that most of them (24.24%) are primarily related only to the goal "Industry, Innovation, and Infrastructure". The most represented single SDG among 83 supply-chain-related studies was identified as "Decent work and economic growth" which was related to supply chain studies only.

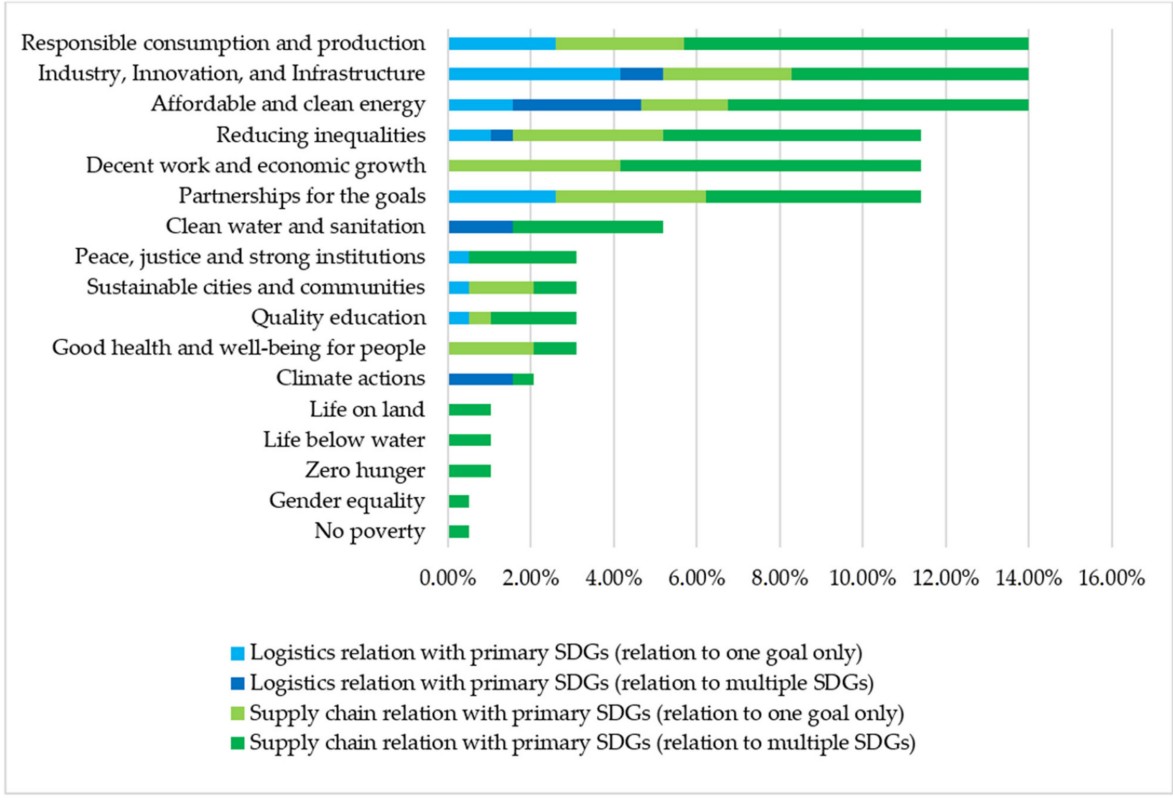

**Figure 4.** The identified primary orientation of logistics and supply chain studies towards specific UN Sustainable Development Goals.

### 3.4. The Secondary Orientation of Investigated Studies towards Specific SDGs

Because some studies' primary and secondary orientations differ, a separate analysis of the secondary orientation of studies towards separate SDGs was performed. Even though studies can be inter-related with multiple SDGs they were divided based on primary and secondary orientation. If the study's focus is related to more SDGs, these were defined as a primary orientation. Suppose the emphasis (content) of one study is related to one or more SDG and the same study is related to a specific sector, which cannot be identified as a primary focus of the study, this was identified as the secondary orientation of the paper. The primary focus of one study is identified to be, for example, "Responsible consumption and production", but since the study investigates the impact of sustainable consumption in different urban communities of the future, its secondary focus is related to "Sustainable cities and communities". Secondary orientation was therefore analyzed separately. In secondary orientation, 9 out of 17 SDGs were identified 125 times in 116 examined studies, meaning that fewer studies had a secondary orientation defined. Orientation towards multiple goals was rare here. In 33 logistics-related studies, secondary relations to SDGs were identified 37 times, and in 83 supply-chain-related studies 88 times (see Figure 5).

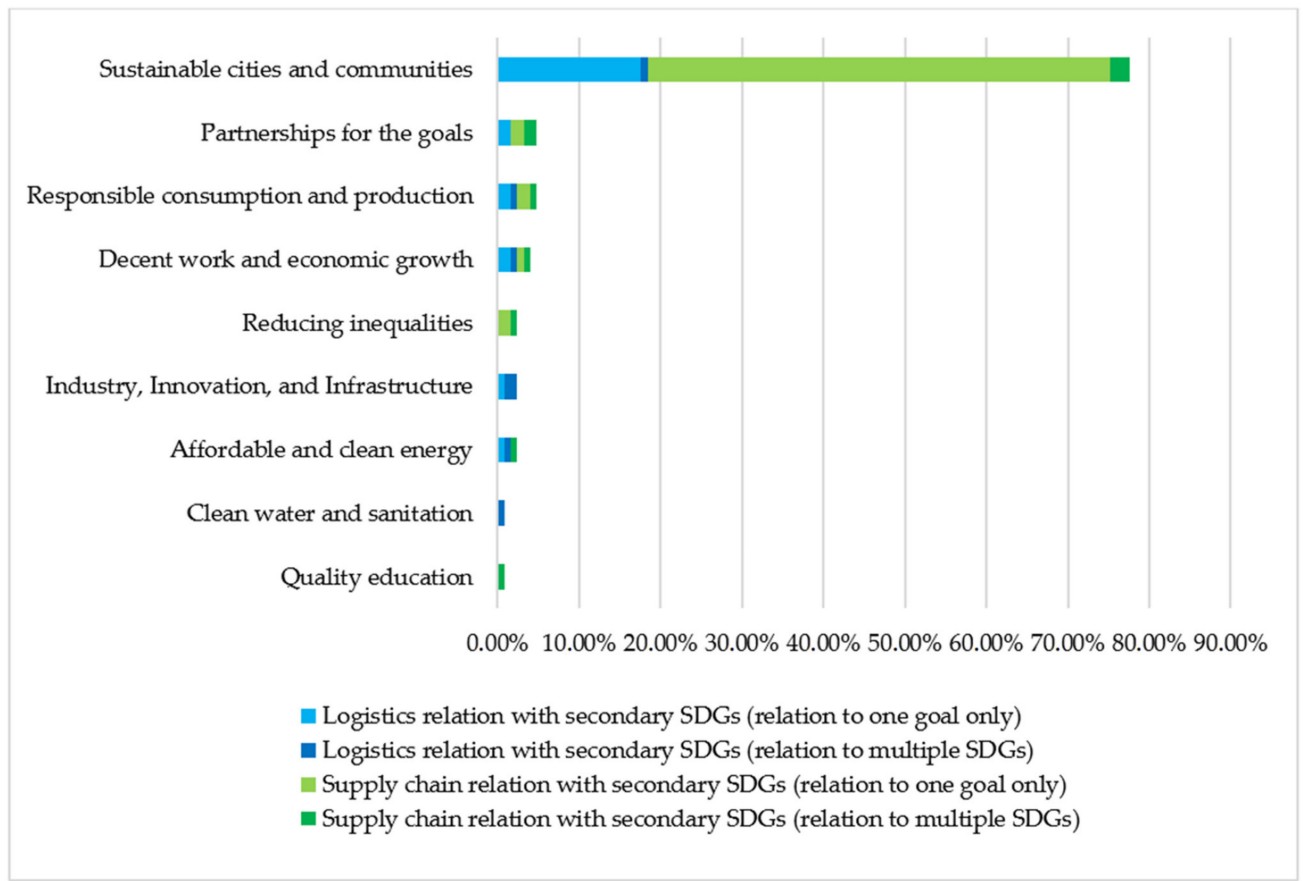

**Figure 5.** Identified secondary orientation of logistics and supply chain studies towards specific UN Sustainable Development Goals.

One SDG prevailed strongly among all so it can be concluded that secondary orientation is focused mainly towards: "Sustainable cities and communities" (97 cases). It stands out in logistics (23 times) and supply chain (74 times) related studies. In total, 80.17% of studies' secondary orientation can be linked with "Sustainable cities and communities" as the only secondary orientation. This means that logistics and supply chain studies are strongly related to this SDG. Even though not among primary priorities, most of the studies addressed sustainable cities and communities as a secondary orientation, which

can lead to the conclusion that this might be one of the top future priorities not just for scientists but also for politics, supply chain, and logistics managers as well as different public stakeholders.

### 3.5. Bibliometric Analysis of Reviewed Literature Network Connectivity

Several different bibliometric analyses were performed using VOSviewer according to the Scopus database: (1) Overview of the number of studies which have at least five quotes regarding citation; (2) Network connectivity of all keywords which were repeated at least twice (with nine clusters); and (3) Network connectivity of keywords which were repeated at least twice with all three main topics/keywords of this research: sustainable development, logistics, and supply chain (also upgraded with two standard weight attributes).

By far the most recognized and cited is currently the study of the authors Brandenburg et al. [54] with 609 citations followed by the study of Isaksson et al. [38] with 65 citations defining a big gap among most cited and all others. In Figure 6, a visualization of the citation's network with at least five quotes is presented. As expected, studies published before 2018 dominate. Newer studies will probably gain more citations in the following years however, due to a significant increase in the number of published papers in 2018, 2019, and 2020 future citation networks could be more dispersed instead of focused mainly on one widely known study.

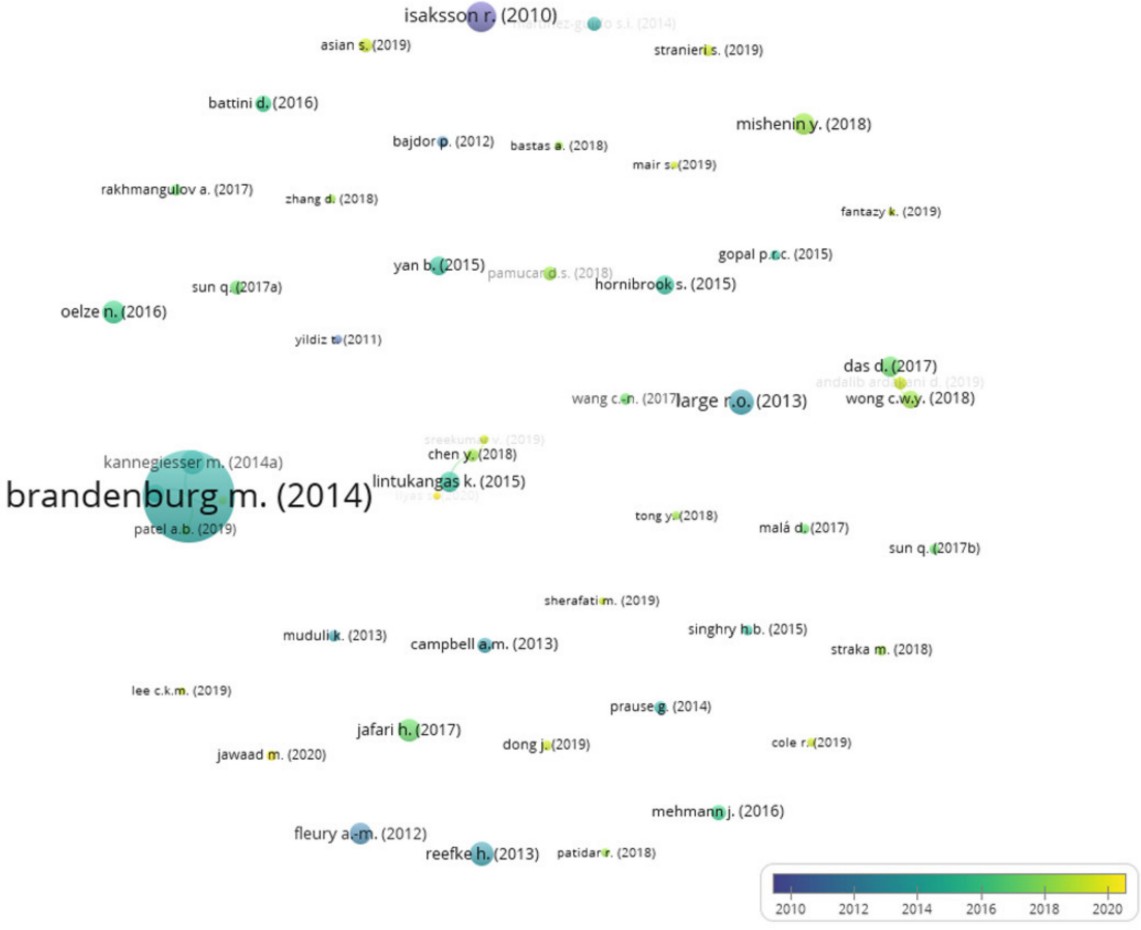

**Figure 6.** Visualization of citation network (citation mapping).

More related to the study content orientation was an analysis of keyword connectivity. VOSviewer identified 809 keywords, and when the minimum number of occurrences was set to repeat at least twice, 139 keywords met the threshold. The most frequently identified keyword was "sustainable development", which occurred in 69 cases followed

by "sustainability", which was repetitive in 32 cases. Till seventh place follow supply chain management, planning, logistics, supply chains, and sustainable supply chains. Figure 7 shows the network connectivity's visual presentation regarding minimal keyword occurrence with all identified keywords. The circle and letter sizes point to the number of occurrences of the keyword, and the links show which keywords appear together in studied literature. Based on VOSviewer analysis, nine clusters of keywords were formed that had common means and content. The most prominent clusters that occur are (1) the red one (named "*Sustainable supply chain management*"), which has in common 29 keywords out of 139 that repeated at least twice. The keyword focus is sustainable development, supply chain management, and green supply chain management. (2) The second, and (3) the third most occurred clusters are the green one (named "*Food supply and supply chain decisions*") and dark blue (named "*Smart and resilient communities*"). The green cluster is formed by keywords such as supply chains, food supply, and decision-making. The dark blue cluster is defined by keywords such as, for example, smart city, urban development, and climate change. Both have 20 common keywords. (4) The fourth is the yellow cluster (named "*Green logistics*") which has 19 common keywords. This cluster is constructed on keywords such as logistics, green logistics, and environmental protection. (5) The fifth cluster is purple, which has in common 15 keywords (named "*Environmental optimization of supply*") like sustainable supply chain, environmental sustainability, and optimization. Slightly less represented are clusters visualized with the following colors: light blue (6), orange (7), brown (8), and pink (9) that have altogether 36 keywords in common; therefore, their connection is slightly less significant.

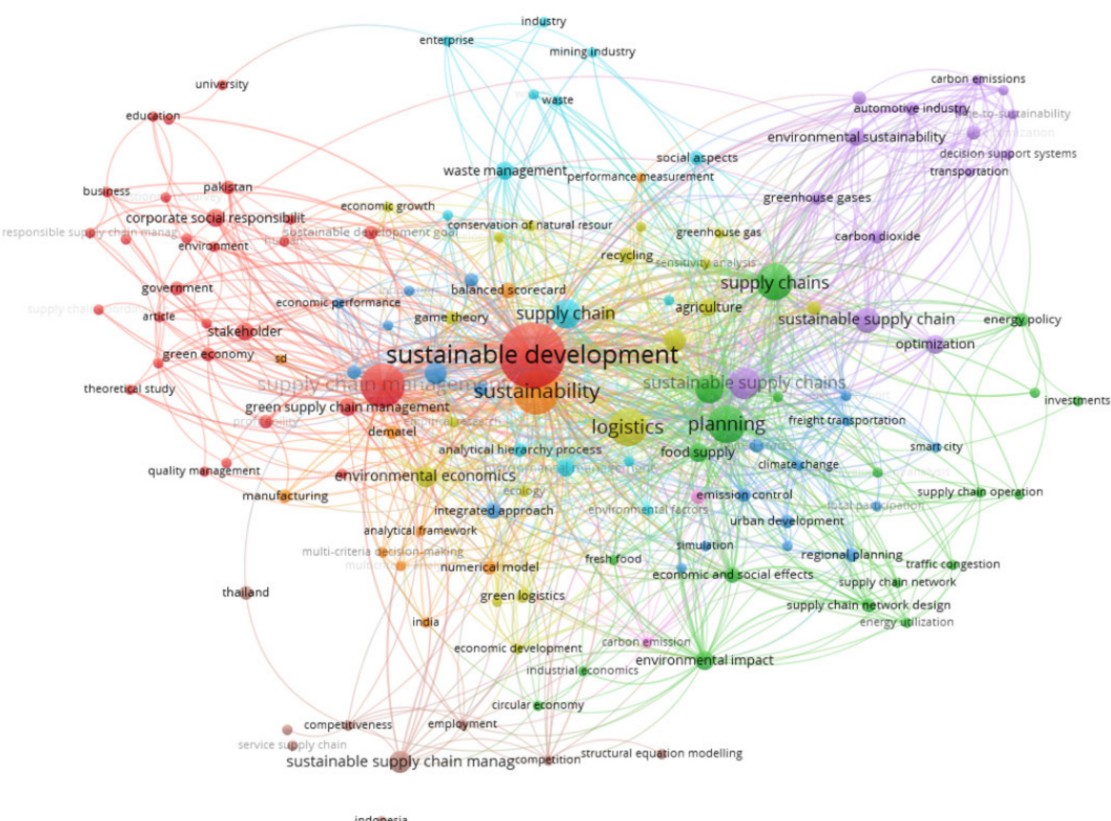

**Figure 7.** Visual presentation of keyword co-occurrence network connectivity with nine identified clusters.

The next analysis was made to assess the network connectivity of keywords that were repeated at least twice within all three main topics of this research: sustainable development, logistics, and supply chain with the same nine clusters mentioned in the previous paragraph. This part of the study was also upgraded with two standard weight

attributes regarding VOSviewer: (1) links attribute and (2) the total link strength attribute. Mentioned attributes indicate the number of links of an item with other items and the full strength of the ties of an item with other items [150]—in this case, the item presents other keywords. The total number of links in this analysis is 1438, and the total number of link strength attributes is 2167. A visualization of the relations of "sustainable development" is revealed in Figure 8. The keyword sustainable development is part of the red cluster ("*Sustainable supply chain management*") and is related to 134 links and has 419 link strength attributes. It was also possible to find keywords for sustainable development: "sustainable development goal" and "sustainable development goals" in the same red cluster. In Figure 9, as previously presented, connectivity is separately shown for relations focusing on logistics (Figure 9a) and on supply chain (Figure 9b). The keyword "logistics" is part of the yellow cluster related to 63 links and having 128 link strength attributes. Regarding the keyword "logistics", it was also possible to find green keyword logistics in the same yellow cluster. On the other side, the keyword "supply chain" is part of the light blue cluster and is related to 36 links and has 57 link strength attributes. It should be emphasized that the keyword "supply chain" was also found in different forms (e.g., green supply chain management, responsible supply chain management, supply chain sustainability, supply chains, green supply chains, sustainable supply chain management, etc.). In total, the keyword "supply chain" appeared in 17 different forms, which were part of 6 clusters (red, green, dark blue, purple, light blue, and brown).

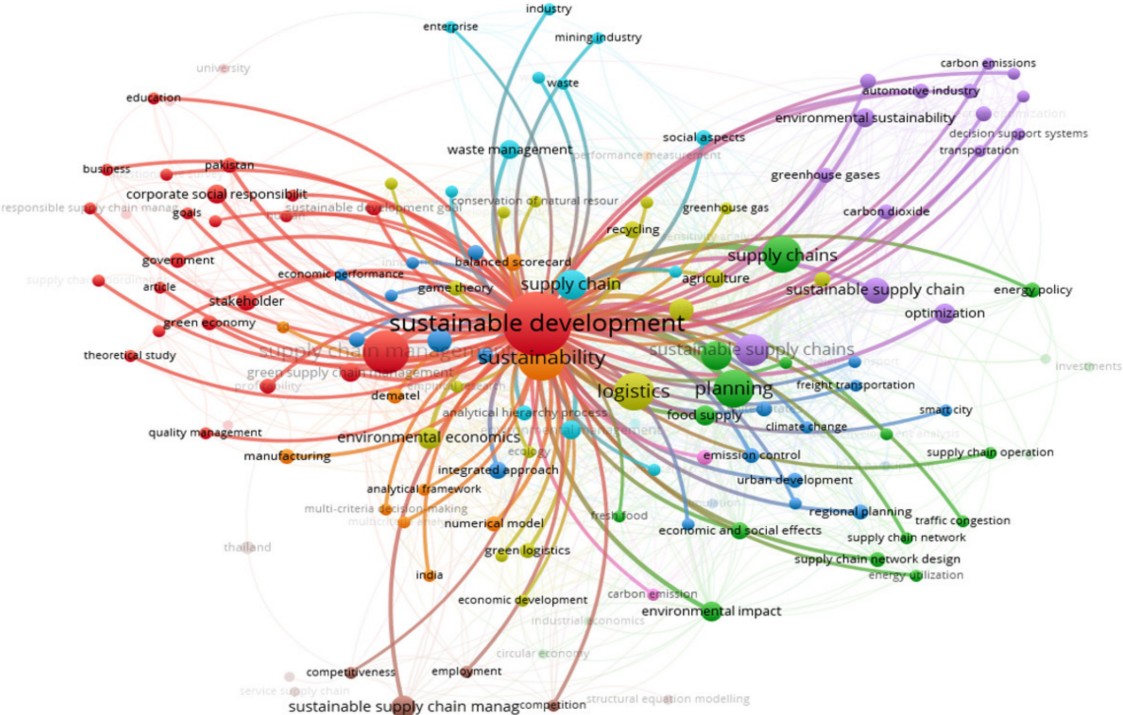

**Figure 8.** Visual presentation of network connectivity for "sustainable development" according to relations with logistics AND supply chains divided into nine clusters.

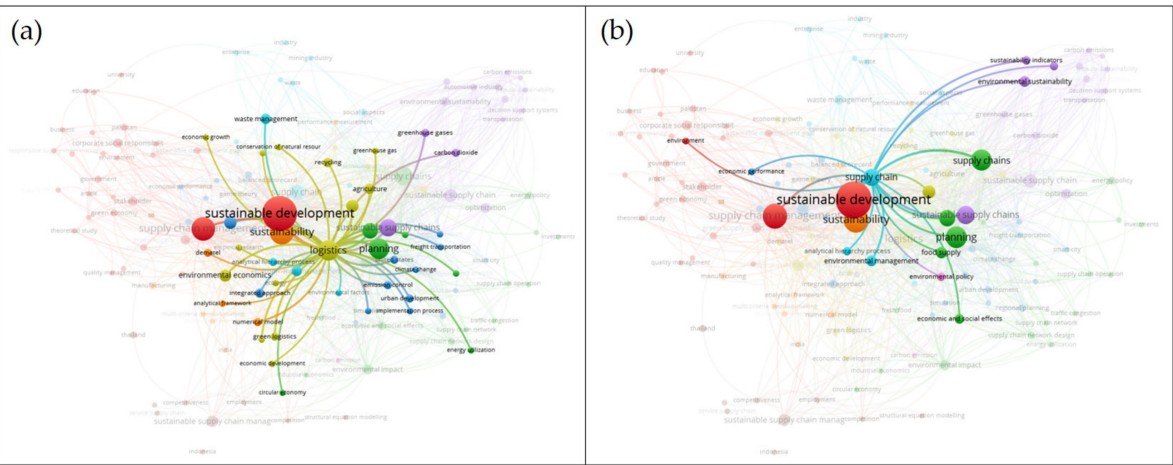

**Figure 9.** Visual presentation of network connectivity (**a**) for the keyword "logistics" and (**b**) for the keyword "supply chain", both according to the criterion repeated at least twice with two search conditions—1st: TITLE (sustainable AND development AND logistics) 2nd: TITLE (sustainable AND development AND supply AND chain).

## 4. Discussion

The chosen research field is gaining international attention and is identified as extremely topical. Regarding the number of published scientific papers, they can be seen mostly from 2018 onwards. In any case, the fact that the year 2020 was taken into research for only the period from January to the beginning of October should be emphasized. It is expected that the trend of interest in this area will continue to grow in the following years, and relating SDGs with specific sectors will add value to research and policy implications. Currently, national studies are strongly represented and prevailing because there are many analyses on studies in only one country or even a city. However, it was evident that the importance of integrating the international perspective of research areas is growing. Internationally oriented studies were well represented already by 2014. After that, an unexpected decline was perceived and, from 2018, the increasing importance of the international perspective is seen again (48.28% of international studies). The international perspective and comparison among different countries, cultures, and nations can bring additional insight to achieving SDGs even more efficiently by applying specific approaches in different environments as proposed by Knez et al. [151] for country-specific policies promoting zero-emission vehicles in three different EU Member States.

The field of research needs to be considered in multiple sustainability dimensions. On one side, 81.90% of all studies were related to at least two, and 47.41% were related to all three sustainable development dimensions. The potential in sustainable supply chain competitiveness is seen especially if it is operationally, environmentally, and socially successful [72]. When analyzing sustainability dimensions, all the dimensions considered individually or together, most studies are still primarily related to the economic dimension (83.62%), meaning that the economic perspective is still the primary force that designs supply chains. Despite that, the increasing integration of the environmental dimension (detected at 77.59%) shows that environmental issues might not yet be equally crucial to economic ones but can be foreseen to be on the way to maybe even succeeding them. Reasons for this could be seen in the political focus towards decarbonization (recently also in China and USA), tackling climate change on a global as well as business level, the post-COVID transition to a greener industry (e.g., European Green Deal), and increasing consumer environmental awareness (e.g., controversial palm oil supply chain and avoiding of palm oil by numerous responsible users). The least represented was the social dimension (68.10%). The social dimension is understudied and will follow the environmental dimension in priority in the next five years [152]. It is already increasing in importance among socially responsible companies that assess and monitor social responsibility not just for themselves but also for their Tier-1, Tier-2, Tier-n sub-suppliers, recently being practiced

in food supply chains and some clothing manufacturers. However, the social dimension is not yet integrated into the supply chain and logistics sector sustainability. Greater focus on supply chain reliability, responsiveness, and avoiding supply disruptions with shorter supply chains might also increase supply chain transparency to a higher level and enable more reliable monitoring of supply chain sustainability metrics.

Being confirmed by all United Nations member states [115], the 17 SDGs offer opportunities that current and future generations impose on current leaders, governments, and the business world, to enable long-term sustainable prosperity. Since sustainability dimensions need to be monitored comprehensively in different parts of the globalized processes, supply chains, as well as globalized logistics processes, get an important role as they become the initiators of the necessary changes and at the same time become the operators of interconnected, sustainable systems based on SDG application [133]. This is one reason why new research in the area of SDGs with a purpose and goal to find new insights, regardless of the study area, is crucial. This research focuses on the logistics and supply chain sector but might also be related to other cases, for example, stakeholder perception towards hydrogen energy [153], electric vehicles [154], etc.

When examining logistics and supply chain's primary and/or secondary relations to SDGs, studies are often related to multiple SDGs, meaning that they are interdisciplinary. Both in logistics- and supply-chain-oriented papers, specific SDGs can appear alone or in combination with other goals in one individual study; however, there are slightly more logistics studies related with one SDG only, contrary to supply chain studies that are more often related with multiple SDGs. This is valid also from the supply chain complexity perspective since supply chain partners need to adapt to multiple variables and partner goals. The most represented are three primary SDSs: (1) Responsible consumption and production, (2) Industry, Innovation, and Infrastructure, and (3) Affordable and clean energy. This is again in accordance with the proposed long-term sustainable development paradigm, focused on cleaner production, circular economy, and reduced consumption; decarbonization of the energy sector and promoting innovative solutions for the transition towards a more sustainable future, especially in the EU but partly also in China and the USA as the most important global players shaping the sustainable development agenda. Within secondary SDGs, the priority is identified to be "sustainable cities and communities" which can be related to 80.17% of all studies. This is not surprising since urban living is becoming a prevailing way of life. Projections of 68% of the global population living in cities in the future by 2050 [155] require sustainable cities and communities, and this critical global challenge is appropriately addressed as an indirect, secondary focus of most of the studied papers. This goal also appears as part of the combination with other goals, but in smaller examples than in primary goals (even when it appears, "sustainable cities and communities" is again in the foreground). Applicability of the results in practice is seen especially in gaining better insight on which sustainability priorities matter in logistics and supply chain research at the moment and using these finding to foresee future trends related to the development of sustainable logistics and supply chains as supportive sectors, for example, for the development of sustainable cities, infrastructure, sustainable food supply, energy transition. Since supply-chain-related studies are more interconnected with all three sustainability dimensions simultaneously as well as with the SDGs than logistics studies, it could be speculated that just individual logistics companies' activities are not enough for achieving sustainable development. The sum of a comprehensively active network of supply chain partners leads to achieving a more sustainable future and specific SDGs.

Bibliometric analysis has revealed that several authors usually produce the studies as it is the current practice due to a higher degree of interdisciplinarity in sustainability research. The most cited papers were published in the high-ranked Journal of Cleaner Production, which has been rapidly growing in recent years. This is in accordance with the best-ranked primary SDG orientation in this study—"Responsible production and consumption" directly related to the journal's scope.

"Sustainable development" and "sustainability" network connectivity analyses revealed that all three main topics of this research: SD, logistics, and supply chain, are connected with nine clusters. The majority of connections are focused on "*Sustainable supply chain management*", "*Food supply and supply chain decisions*", "*Smart and resilient communities*", "*Green logistics*", and "*Environmental optimization of supply*". These research results can be a valuable insight for forecasting future trends in the development of logistics and supply chain managerial decisions and business focus since strong relations with sustainable development priorities can be seen as one of the top business management as well as global policy priorities. It does not matter if SD dimensions or SDGs are studied, a clear trend is seen. On the one side, the current trend shows that all SD dimensions together are quite well represented with social perspective slightly lagging behind and on the other side, some SDG relations are much more highlighted than others. Monitoring current trends is a valuable tool for change management, enabling SD implementation in all types of logistics processes and supply chain organizations more efficiently and on-time. The practical perspective of research findings is the social implication of valuable information for managers in the logistics and supply chain segment and will enable them to implement new insights into future business operations. It is supposed that along with helping managers to make specific decisions on supply chain strategy and the strategic focus of their companies and to avoid risky situations related to unsustainable business practice by following top SDG priorities, this paper also provides a new understanding of more sustainable business logistics and develops social wellbeing worldwide with equalizing the importance of all three pillars of sustainability.

## 5. Conclusions

This paper aims to provide a comprehensive and systematic analysis of sustainability focus and orientation in supply-chain- and logistics-related studies in the last decade to clearly and unequivocally define which sustainable development UN goals and sustainability dimensions stand out.

After 2018, there was a notable increase in studies (especially international ones), proving that SD became an inevitable, relevant, and topical trend on the worldwide level in contemporary life. Detailed insight into the current SD situation enables predicting future trends in logistics and supply chains.

Overall findings indicate that further activities about SD in practical and theoretical research areas should show attention to multiple SDGs. Some of the primary goals stand out, especially: "*Responsible consumption and production*", "*Industry, Innovation, and Infrastructure*", and "*Affordable and clean energy*". Prioritization enables logistics companies to understand the meaning of individual goals and focus on the most important as well as on their impact on the companies' business structure. Among secondary SDGs "*Sustainable cities and communities*" clearly stands out, showing logistics companies that more attention should be given to smart and sustainable cities. It is also not surprising that all three SD dimensions are very well represented (individually and collectively) since SD is multifaceted. The economic dimension stands out in terms of occurrence.

Nevertheless, this paper is not without its limitations, and this could definitely serve as encouragement and also an exceptional opportunity for future work and discovering new insights in the area of supply chain and logistics. The progress of this paper enlights sustainability focus and orientation. Because implementation of the sustainability concept remains the biggest challenge globally [156–158], it is, consequently, even more important to understand which goals and dimensions special attention should be given to. Focusing on investigating relations to SDGs and sustainability dimensions on a micro-level in real-time, for example, based on daily operations, short-therm plans, logistics processes, strategic and operational management, would be very beneficial for a sustainable future for all. Therefore, especially, the mitigation of supply chains to climate change, the impact of clean energy in logistics processes, responsible consumption and production, logistics in smart cities, etc. should be further investigated to provide new knowledge for the future.

**Supplementary Materials:** The following are available online at https://www.mdpi.com/2071-105 0/13/6/3280/s1.

**Author Contributions:** Conceptualization (S.L. and M.O.); Formal analysis (S.L.); Funding acquisition (M.O. and D.K.-T.); Methodology (S.L. and M.O.); Visualization (S.L.); Writing—original draft (S.L.); Writing—review and editing (M.O. and D.K.-T.). All authors have read and agreed to the published version of the manuscript.

**Funding:** This research received funding for publishing in a journal from the University of Maribor, Faculty of Logistics (Slovenia) and the Czestochowa University of Technology, Faculty of Management (Poland).

**Institutional Review Board Statement:** Not applicable.

**Informed Consent Statement:** Not applicable.

**Data Availability Statement:** Due to the large volume of data, this paper's database is submitted as Supplementary Materials in a separate file which is a supplement/extension of this paper.

**Conflicts of Interest:** The authors declare no conflict of interest.

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
