# Peer review of "Sustainability Orientation and Focus in Logistics and Supply Chains"

_sustainability, doi:10.3390/su13063280_

Round 1
Reviewer 1 Report
The topic of the paper is interesting and up-to-date. The main problem described in the paper is very and paper is generally well written. It is theoretical paper based on relevant and up-to-date literature very big amount of literature.
But it has some flaws:
- Maybe it would be useful to add some research questions.
- Please describe the links between the research gap and the goal of the paper and research question.
- Add some social implications to the paper.
Author Response
Thank you for reviewing our manuscript and presenting your perspective and proposals for potential improvements of it. The English text has already been proofread, however due to significant changes and improvements, it was again additionally checked. Regarding your comments/proposals we are explaining point by point that:
- “Maybe it would be useful to add some research questions. Please describe the links between the research gap and the goal of the paper and research question.”
Reply: We added research questions in the last paragraph of 1.3 subchapter of the introduction. We also changed other parts of this subchapter a little bit to better emphasize the gap and its relation with manuscript goals and novelty of the work. Regarding the proposal of reviewer #2 we have also shortened paper title and change the abstract as well as divide introduction on three parts (1.1 Background, 1.2 Literature review and 1.3 Gap, goals and research questions.
- “Add some social implications to the paper”.
Reply: We have added some social implications and managerial insights in the last paragraph of discussion and some in conclusion. We especially focused on the practical perspective and additionally defined proposed direction of the future study topics.
Reviewer 2 Report
The paper is about redefining sustainability. However, there are the following comments for the study.
Comments:
- The first sentence of the abstract is wrong. It must be corrected properly.
- The title must be shortened. The significant findings should be added in the last of the abstract.
- The abstract must be rewritten carefully to show the necessity of this research, novelty, contribution, and significant findings.
- The introduction and literature review should be separated. Several significant studies regarding sustainability are missing: Joint effects of variable carbon emission cost and multi-delay-in-payments under single-setup-multiple-delivery policy in a global sustainable supply chain, A sustainable development framework for a cleaner multi-item multi-stage textile production system with a process improvement initiative
- The introduction should contain the model's contribution with the exact research gap. The research gap should be adequately explained.
- See the paper mentioned above and make the author contribution table like this paper.
- Write the managerial insights correctly. Do not use we/our/us throughout the study. Replace those sentences in other ways.
- Make the comparative study with the exiting research to show what is the novelty in this model.
- The proper extension of this paper should be provided.
Author Response
Due to long reply I attached separate file as a response to reviewer no. 2

Round 2
Reviewer 2 Report
The paper has improved a lot